# Hydrothermal Conversion of Food Waste to Carbonaceous Solid Fuel—A Review of Recent Developments

**DOI:** 10.3390/foods11244036

**Published:** 2022-12-14

**Authors:** Moonis Ali Khan, Bassim H. Hameed, Masoom Raza Siddiqui, Zeid A. Alothman, Ibrahim H. Alsohaimi

**Affiliations:** 1Chemistry Department, College of Science, King Saud University, Riyadh 11451, Saudi Arabia; 2Department of Chemical Engineering, College of Engineering, Qatar University, Doha P.O. Box 2713, Qatar; 3Chemistry Department, College of Science, Jouf University, Sakaka 72388, Saudi Arabia

**Keywords:** hydrothermal carbonization, food waste, hydrochar, reactor

## Abstract

This review critically discussed recent developments in hydrothermal carbonization (HTC) of food waste and its valorization to solid fuel. Food waste properties and fundamentals of the HTC reactor were also covered. The review further discussed the effect of temperature, contact time, pressure, water–biomass ratio, and heating rate on the HTC of food waste on the physiochemical properties of hydrochar. Literature review of the properties of the hydrochar produced from food waste in different studies shows that it possesses elemental, proximate, and energy properties that are comparable to sub-bituminous coal and may be used directly as fuel or co-combusted with coal. This work conclusively identified the existing research gaps and provided recommendation for future investigations.

## 1. Introduction

Managing the increasing volume of food waste worldwide has become a major challenge. In addition, the energy crisis in developing nations of the world is currently increasing at an alarming rate and the need to proffer solutions is not only expedient but must be of high priority. On average, developed countries generate 100–170 kg of food waste per capita per year, corresponding to more than twice of that produced in developing countries [1]. Research has shown that liquefaction, microwave heating, pyrolysis, and slow pyrolysis can be used to convert food wastes to valuable solid products. However, unlike hydrothermal conversion and anaerobic digestion processes, there is an additional requirement of pre-drying the wet food wastes [2].

Hydrothermal carbonization (HTC) refers to a conversion of biomass into an energy-densified or carbon-rich char product. It involves hydrolysis, dehydration, polymerization, and carbonization reactions taking place within moderate temperature (between 180 and 260 °C) and pressure (between 35 and 55 bar) ranges [3]. The hydrothermally developed solid product could be used as a fuel, an adsorbent, and a catalyst, while liquid product with high acid and phenol concentrations has very limited applications. Carbon dioxide (CO_2_) is the major constituent of hydrothermally developed gaseous product, along with hydrocarbons in trace amount.

During recent studies, the biomass substrates used as feedstocks in the HTC process include sludge from dairy processing [4], sewage sludge from wastewater treatment plants [5], algae [6], and municipal waste [7]. These wastes due to industrialization and urbanization are generated in huge amounts. In addition, agro-waste feedstocks (such as giant bamboo, coffee wood, eucalyptus, and coffee parchment) have also been used as feedstock in HTC processes [3].

Another suitable feedstock for HTC is food waste, which is often generated in large quantities and from different sources along the food supply chain. In developed countries, the massive amounts of household food wastes generation was a reason for United Nation to set a goal with an aim to cut the extensive food loss to half by 2030. The indiscriminate dumping of food waste in the open field and in landfills is currently creating health and safety concerns in many countries as this activity not only deteriorates the environment but is a serious cause of global warming. Nonetheless, food waste could serve as a useful feedstock resource for the production of value-added products such as carbonaceous solid fuel via HTC.

The biggest generators of food waste are fruit juice industries, school canteens [8], and dining halls [7]. Food waste is mainly composed of cellulose, hemicellulose, and lignin by chemical structure and proteins, lipids, and carbohydrates by classification or category. It is an organic compound containing trace amounts of elements such as nitrogen, phosphorous, and potassium [9]. Food waste contains moisture which when subjected to HTC under increased temperature and pressure serves as an organic solvent owing to its decreased dielectric constant. Thus, subjecting food waste to pyrolysis, gasification, and combustion may not be economically feasible as a result of the energy-intensive requirement for vaporizing the inherent moisture.

Previous works revealed that HTC of food waste to carbonaceous solid fuel is feasible in terms of technical and economic requirements. For instance, Gupta et al. [10] found that HTC of food waste (200 °C, 1 h) produced a carbonaceous solid fuel with heating value of ~30 MJ/kg. Energy required to initiate the combustion reaction for solid fuel generation was about 54% less compared to raw food waste. Similarly, HTC of pineapple and watermelon peels produced hydrochars with respective yields and energy contents ranging between 25 and 69% and 17 and 22 MJ/kg [11].

There are sufficient studies on the application, mechanism and influencing parameter of the hydrothermal process for solid fuel production from a diverse and extensive list of organic wastes. While Zhuang et al. [12] reviewed the current advances in the practical applications and benefits of HTC, Melikoglu [13] discussed the food waste utilization techniques such as composting, anaerobic digestion, fermentation, and thermochemical conversion and concluded that thermochemical conversion through valorization could endanger the environment and the supporting costs may far outweigh the economic benefits. A review by Pauline and Joseph [14] emphasizes the influence of HTC process parameters for wood, animal residues, sewage, and municipal solid waste conversion besides evaluating the composition and yield of hydrochar. However, these reviews did not specifically focus on food waste; neither its properties nor critical process parameters for HTC conversion of food waste to hydrochar were discussed.

Thus, to correctly optimize the required parameters for high mass and energy yield solid carbonaceous fuel generation, comprehensive analysis of properties and process parameters on HTC of food waste is essential. To fill this gap in the literature, this review focuses on the properties of food waste and food waste-derived hydrochars. In addition, the HTC reactor fundamentals and effect of process parameters were elucidated and summarized. Finally, the existing research gaps in this field were discussed along with corresponding recommendations.

## 2. Methodology

While avoidable waste can be associated to that generated in phases of distribution, marketing, and consumption of food, this paper conceived to select and examine research dealing with unavoidable food waste conversion (such as prepared, unconsumed, and expired foods from various sources) to solid fuel. Efforts are made to understand each and every aspect of HTC of these types of food waste in relation to their properties, HTC process parameters such as process temperature, heating rate and time, and water–biomass ratio that influences the physico-chemical characteristics of the hydrochars and ultimately their combustion kinetics. Representative publications around these themes from recent research works indexed in *Scopus* and *Web of Science* (WoS), which goes back 5 years (2018–2022), are reviewed and analyzed. The keywords selected for searching the databases include: hydrothermal carbonization of food waste, hydrothermal carbonization of restaurant waste, combustion kinetics of food waste.

## 3. Food Waste

### 3.1. Availability

Socioeconomic development is a major cause behind an alarming rise in both unwanted and unintentional release of food which ends up aswaste from different sources. The quantity, quality, and availability of the food waste vary from country to country and location to location. Recent surveys conducted have shown that food waste is available in large quantities in both developing and developed countries. A survey conducted by Herzberg et al. [15] in Germany revealed that the food waste amount per household was approximately 37.8 kg. Upon extending the trend nationwide, about 3.7 million tons was estimated to be the quantity of domestic food wasted during the period and duration of the survey. Another field survey undertaken by Silvennoinen et al. [16] indicated that about 17.5% of cooked food was wasted, of which 2.2, 11.3, and 3.9% were kitchen waste, serving losses, and table scraps, respectively. The average quantity of cooked food waste was 449 g for each category, of which 78 g was eventually discarded in refuse dumps. The study revealed that a significant portion of serving food losses chiefly contain meat, fruits, and vegetables.

Garcia-Herrero et al. [17] undertook a field survey in Spain. The survey’s outcome uncovered that ~20% of the national food production is estimated to be lost or wasted. It was found that the quantity of domestic food waste generated annually was 88 kg/person. Additionally, perishable fruits and vegetables formed almost half of the bulk food waste generated. Overall, the total quantity of the food waste generated annually was roughly 65 kg/cap and represents roughly a fraction of 0.6 of food waste generated. Next in order of the quantity of waste food generated were cereals, which was a fraction of 0.2 of the total food waste generated.

In the European Union, Caldeira et al. [18] provided data on food waste generated for different food categories. The major chunks of food waste observed in the survey were fruits and vegetables. Apart from these, eggs and fish were found in sparse quantities. Comparatively, fish formed the largest share of aggregate food waste relative to fruits and vegetables. The data collected also showed that roughly 129 tons of food waste were generated and about 46% of this amount was generated at the consumption level. At the primary production, processing, and distribution stages, the report revealed that 25, 24, and 5% of food waste was generated, respectively.

Li et al. [19] conducted a field survey for 207 residential houses in Shandong province in northern China. On average, the food waste generated per meal was 8.74 g/cap, of which about 0.9 and 0.5 fractions of the food waste were plant- and vegetable-based, respectively. In China, numerous factors such as household income, size and age of the families, and the food varieties obtainable were found to greatly influence the amount of food waste generated in Chinese households located in the rural area.

Abdelaal et al. [20] carried out a field survey at various food-selling locations around a university campus in Qatar with a view to quantitatively and qualitatively analyzing the food waste generated in the area. The aggregate amount of waste generated daily was estimated to be 329.5 kg. They concluded that surplus food production was a major reason for unpreventable volumes sof food waste generation. The categories of waste in the food waste mix were predominantly fruit and vegetable scraps and animal processing waste such as that from meat pruning.

The outcomes of all these surveys give credence to the report that no less than a quarter of global food production gets lost even before being processed in the households for consumption. In quantitative terms, the global food waste is reported to be in the tonnage of 1.3 billion, and this is not without severe negative impact to the environment. Recently, valorization of these food wastes into value-added products using thermochemical techniques is dominating the research space, as seen in published works. Unlike conventional biomass raw materials, food waste is somewhat unexplored despite the fact that it is readily available in abundant quantities in both developed and developing countries.

### 3.2. Physicochemical Characteristics for Energy Applications

Chemical composition (cellulose, hemicellulose, and lignin content), heating value, fuel ratio, ash content, and oxygen-carbon ratio (O/C) are some of the important properties or tools that can be used to access the potential of unavoidable food waste as solid fuel. The fuel ratio, a fixed carbon to volatile matter (FC/VM) ratio, may be used to rank fuel source potential of food waste and the substitution to hydrochar as an alternative coal fuel. A high fuel ratio is indicative of excellent fuel quality or a high ignitability index [21]. Ash is another important property and can be used to access the fuel source potential of food waste or hydrochar. The analysis of the ash content is necessary to access the tendency of slag and foul formation when used as fuel in boilers or combustion engines. The deposition of slag and foul on the walls of the boilers or engines contributes to the reduced efficiency of the boilers. For instance, ash comprising Na_2_O and K_2_O reacts with boiler surfaces to produce low-melting point compounds.

The chemical and elemental (carbon-C, hydrogen-H oxygen-O, sulfur-S, and nitrogen-N, etc.) compositions of food waste and hydrochar are used to determine the fuel quality of food waste or the derived hydrochar. Food waste rich in lignin is desirable to produce solid fuel with high yield because of its relatively high thermal stability compared to cellulose and hemicellulose [22]. As a further assessment, the O/C ratio of the food waste determined from the result of the elemental composition not only accesses the stability of the produced solid fuel but also identifies/distinguishes the fuel from other carbonized materials. Lower the O/C ratio, the higher the heating value of the produced solid fuel.

Food wastes comparatively contain high moisture content relative to other waste fractions found in municipal wastes, and as a result, combustion or conversion into useful products requires a great energy input. Hitherto, the relatively high moisture in food waste, especially in fruits and vegetables, causes the release of dioxins when burnt with other organic material [23], thus endangering human and environmental health. Therefore, high moisture content is not a worthy characteristic of food waste because more energy is required to eliminate the water molecules present before conversion takes place. The high moisture content in food waste causes it to have a diminished heating value, thus causing it to possess very low energetic quality.

Food waste from different sources or at different generation stages may have different characteristics. Table 1 [24,25,26,27,28,29,30,31] presents the characteristics of different food wastes reported in some studies. One of the major shortcomings of these investigations is the use of comparatively minute masses of the food waste. Another may be the inconsistencies in food categories and quality of food waste. It is advised that a sizeable amount of food should be analyzed in order to strengthen the energy model for the food waste. Discussion of these characteristics will provide valuable insight into appropriate techniques or processes that can convert food waste into carbonaceous solid fuel.

A careful search in the literature showed that food wastes used during different investigations had moisture content ranging from 1.59% to as high as 74% (Table 1). This range is far higher than the range specified for coal (0.8 to 2.7%) [32]. If food waste’s pelletization is to be contemplated, its moisture is not expected to exceed the range of 6–12% as specified in the literature depending on the category of the food waste [33]. According to Nayak and Bhushan, [34], the high moisture content of food waste and its variable chemical characteristics are some of the challenges preventing the successful production of high-yield bioproducts from food waste.

The fuel heating values are negatively correlated with the water content. Thus, high moisture-containing food waste possesses low heating value. In addition, the high elemental oxygen-carbon (O/C) ratio is a reason behind low heating value. In Table 1, food wastes had elemental O/C ratios ranging from 0.41 to 1.06, while the heating value ranged between 10.54 and 25.32 MJ/kg. The high heating value (HHV) of animal-based food waste (Table 1) is probably as a result of a higher hydrogen-carbon (H/C) ratio; upon combustion, it releases combustible gases accompanied by large amounts of energy. Most of the heating values of the food waste used in many studies were not only less than the combustibility index of 23 MJ/kg, but were also higher than the ignitability index of 14.5 MJ/kg [35]. This suggests that most types of food waste may not qualify as alternative renewable fuel without pre-treatment.

Solid food wastes comprise lignin, cellulose, hemicellulose, and extractives [36]. These chemical components are present in hydrolyzed forms in liquid food wastes. These properties are very useful for assessing the energetic potential of food waste for use as renewable fuel raw materials. In the literature, very few studies have analyzed these properties, particularly in real, simulated, or mixed food waste. Food wastes used in different investigations were reported to have lignin contents between 2.68 and 15.61% (Table 1). Pecorini et al. [37] previously reported that the lignin content of real and simulated food wastes ranged from 0.9 to 12%. For cellulose and hemicellulose, the range was from 3.12 to 36.63% and 1.12 to 22.76%, respectively (Table 1). According to Singh et al. [38], the hemicellulose content of food gets decomposed during heating or cooking and is usually negligible in cooked food waste samples. One notable characteristic of food waste is its high cellulose content, which eases its biological and thermal degradability. An investigation was carried out by Pagliaccia et al. [39] to appraise the inconstancy of these three components among cooked kitchen waste, fruit and vegetable scraps, and organic fractions of municipal solid waste. They observed that hemicellulose, cellulose, and lignin were mostly and fully derived from the vegetal and fruit parts of a food waste, while lignin was completely absent in cooked food.

Any food waste having a low fuel ratio (Fixed Carbon/Volatile Matter) is less reactive unless suitable pre-treatment is applied. Higher volatile matter (VM) content suggests that the fuel can be ignited and subsequently oxidized with ease. In the literature, food wastes had fuel ratios ranging from 0.014 to 0.27 (Table 1). This range is far less than the range of 1.46 to 7.10 specified for coal [32]. However, with suitable pre-treatment, food waste may qualify as co-reactant in the combustion of coal. In the literature, it is posited that a relatively high fuel ratio (≥2) may cause incomplete combustion, flammability issues, and thus inefficiency of the heating system [38].

Ash is a residue comprising leftover minerals after the release of energy from food waste. The carbohydrate component of food waste comprises more ash relative to other categories of food waste. The ash content for food waste used during previous studies was in the range of 2.1–13.01% (Table 1), far less than the range specified for coal (7.9 to 15.2%) [32].

## 4. Fundamentals of Hydrothermal Carbonization Process

### 4.1. HTC Reactor Pre-Heating and Reaction Time

In batch HTC reactors, the reaction time is usually an aggregate of pre-heating time, dwelling time, and cooling time, thus it is impossible to ensure uniform temperature in real practice. Many published works have defined reaction in different ways. While some studies have not included the heating time as part of the reaction time [10,40] (other studies [41] have taken it into account. However, Sangare et al. [42] stated that pre-heating time may not be of much significance in continuous HTC reactors where heat-up is faster compared to batch reactors. Furthermore, pre-heating time is expected to influence the chemistry of the reaction because HTC requires varied temperatures which would have extended pre-heating time.

### 4.2. Reactant Media

One of the basic conditions in HTC is the aqueous reaction media in which the feedstock materials are soaked in continuously. Different reaction media have been used in literature for the HTC of food waste [43]. The most common reaction media used in the majority of the studies were subcritical and supercritical water. Water below the its critical point (i.e., 374 °C and 22.1 MPa) is referred to as subcritical water, whereas water above its critical point is referred to as supercritical water. Supercritical water as a reaction medium has several merits compared to regular water because of its unique density, viscosity, ion product, heat capacity, and transport behavior associated with liquids and gases at room temperature and normal pressure, respectively. This results in a substantial increase in the chemical reaction rates. Supercritical water is nonpolar in nature and allows comprehensive dissolution of a majority of carbonaceous organic materials and oxygen [44]. Here, it is worth noting that the dielectric behavior of water varies with temperature. At 200 °C, it is similar to methanol at room temperature; at 300 °C, it is similar to acetone at room temperature; at 370 °C, is it similar to methylene chloride at room temperature; and at 50 °C, it is similar hexane at room temperature. During the HTC process (180–350 °C, 2–10 MPa), the water present in the food waste can directly participate in the reactions as a reaction medium. Apart from subcritical and supercritical water, landfill leachate has been employed as an alternative reaction medium for the HTC of waste biomass. Venna et al. [7] undertook the HTC of food waste and yard waste using landfill leachate as the reaction medium. The heating values of hydrochars produced from food waste (30.2 MJ/kg) and yard waste (22.8 MJ/kg) were comparable to traditional fossil fuels. It was further found that the mass yield of hydochar obtained in landfill leachate medium was comparatively lower than the hydrochar obtained in distilled water reaction medium.

### 4.3. HTC Reaction and Chemistry

Many studies have been carried out using biomass in different reaction media to confirm the reaction chemistry governing their conversion to hydrochar [7,45,46,47]. It is believed that hydrolysis reaction dominates the HTC process in non-polar water media and results in the cleavage of hydrogen bonds. Additionally, it is assumed that ionic reaction dominates the HTC process when the temperature of the reaction media is below critical point (subcritical region) [48]. Ionic reaction is believed to increase the mass yield and determine the carbon distribution of the hydrochar. In most of the studies on HTC, it is reported that between 50 and 80% of the original biomass feedstock were converted to hydrochar, while about 5 to 20% were dissolved in the reaction media, and 2–10% were converted to non-condensable gases [49].

A comprehensive reaction pathway for the HTC of food waste is presented in Figure 1. As depicted, food waste is a complex biomass composed of polysaccharides, lignin, protein, and lipids. The polysaccharide (including starch and sucrose) component of a food waste is converted into their basic unit of glucose and fructose through hydrolysis reaction (100–175 °C). Other reactions taking place co-currently were thermal degradation and depolymerization of the cellulosic, hemi-cellulosic, and lignin structural component of the food waste into water-soluble monomer units (160–280 °C). The protein component of the food waste solubilizes in water via hydrolysis to generate ammonium ions and volatile fatty acids in the liquid phase. Subsequent decarboxylation of the volatile fatty acids generates unsaturated hydrocarbons which combine with the lignin component of the food waste through polymerization to generate products rich in aromatic compounds. They solubilize and subsequently hydrolyze to generate ammonium ions and volatile fatty acids in the system. The volatile fatty acids produced from different routes may undergo decarboxylation to produce unsaturated hydrocarbons. These unsaturated hydrocarbons combine with macromolecules such as lignin to result in aromatization and polymerization. Simultaneously, Maillard’s reactions occurring between the protein and carbohydrate molecules lead to the formation of humus substances.

### 4.4. HTC Reactor Setup

Most reactor setups employed in the HTC of food waste in the literature comprise basic components such as pressurized reactors and nitrogen gas cylinders. For simple laboratory set-ups, pressurized reactors generally made of stainless steel with capped Teflon cylinder of 4 to 5 L total volume and 2 to 4 L working volume are used. A typical HTC reactor set-up is presented in Figure 2. The general problem encountered while using higher working volume is low heat transfer efficiency of the set-up, which increases the time required to complete HTC reactions. Thus, it is imperative to establish the optimal heating rate for the HTC reaction given that a rapid heating would likely introduce a higher temperature gradient between the inner core and the external walls of the reactor. Additionally, the internal heating could be implemented to improve the heat transfer efficiency. The reactors usually employed in most of the studies were conventional single-stage reactors. However, some studies have implemented two-stage reactors comprising hydrolysis and carbonization stages [50]. The experimental results of two-stage HTC revealed that lower energy was required to optimize reaction temperature and time. During two-stage conversion of fecal sludge to hydrochar, a respective hydrolysis temperature and reaction time (where fecal sludge was broken down into lower molecular weight compounds such as oligosaccharides, glucose, and amino acids) of 170 °C and 155 min, whereas a carbonization temperature and reaction time (solid–solid conversion, dehydration and polymerization/aromatization reactions occur) of 215 °C and 100 min were optimized with 25% less energy input than a conventional single-stage HTC reactor [50].

While the capping vessels in some set-ups in the literature excluded a central connection for magnetic or mechanical stirring, most studies used a non-stirred reactor heated with a furnace in a closed-loop system [52]. Stirring the submerged feedstock is of great significance where a higher mass yield of the gaseous product at the expense of the solid product is desired [51]. Moreover, reactor stirring is necessary in order to reduce thermal and concentration gradients inside the reactor. The stirring speed of an HTC reactor used during carbonization ranges between 100 and 700 rpm. Inertial and hydrostatic are the two dominant forces acting on a stirrer. The inertial force and hydrostatic force arise from the rotation of the stirrer and difference in density between the biomass and water in the solution [53].

Most reactors depending on the scale of operation were equipped with a furnace control panel (for on-line temperature control and display), a pressure sensor, type-K thermocouples, manometer, and the agitator control box (to regulate, measure, and display the stirring speed) [53]. Different sizes and shapes of HTC reactors have been used in the literature. However, among them, the most common reactor shape is the cylindrical HTC reactor. A cylindrical HTC reactor is attractive because it provides a large volume required for the reaction. Moreover, there is a uniform pressure distribution due to its comparatively lower surface area than a reactor geometry type of the same volume. A reactor setup for the HTC should be flexible enough to convert any type of biomass waste and minimize reaction time. In recent times, most studies have used microwaves to modify HTC reactors with the goal of maintaining a temperature at or around 180 °C [54].

## 5. Hydrothermal Carbonization of Food Waste for Char Production

### 5.1. Effect of Process Parameters

The yield and chemical properties of hydrochars are largely dependent on the process parameters of the hydrothermal process and feedstock type. The important parameters include temperature, pressure, biomass–water ratio, and heating rate. Among them, temperature exerts a greater influence on the product yields of the HTC. Very high HTC temperatures usually result in low yield of solid hydrochar while favoring higher yields of liquid and gaseous products. Table 2 [2,10,40,41,48,55,56,57,58,59,60,61] presents the properties of hydrochars developed from HTC of food waste under optimum conditions.

### 5.2. Temperature

Temperature affects several characteristics of the final hydrochar produced from food waste. In the literature, the effects of an increase in HTC temperature of food waste on the mass yield of the hydrochar were studied. Microwave-assisted hydrothermal carbonization of food waste digestate was carried out by varying the temperature between 160 and 200 °C. The mass yield decreased with rising temperature from 68.9 to 67.5%. A further rise in temperature up to 260 °C lead to a rapid decline of the mass yield from 67.5 to 58.6% [62]. In another study, HTC of food waste over a wider temperature (between 200 and 300 °C) resulted in a reduced hydrochar mass yield (from 7 to 5.25%) [63]. A similar trend in hydrochar mass yield was also observed for feedstock comprising mainly fruit and vegetable waste, with a small amount of meat waste [60]. The possible cause for a reduced mass yield of the hydrochar in these studies may be the use of subcritical water as the reaction medium. Subcritical water dissociates into H_3_O^+^ and OH^−^ ions at 200 °C owing to the weakening of hydrogen bonding, which results in cleavage of the β-(1–4) glycosidic bonds of the hemicelluosic component of the food waste, further leading to the production of sugar monomers. Subsequent decomposition of the sugars to furfurals and other compounds causes a part of carbon in food waste to be transported into the liquid phase, leading to a reduced mass yield of the hydrochar [64] (Ciftci and Saldaña, 2015). Generally, the mass yield reduction with a rise in temperature is expected because the mass loss rate obeys the Arrhenius equation, which implies that an effect of increasing temperature will result in reduction of the hemicellulose component of the waste.

Liu et al. [40] have found that the effect of temperature on mass yield also varies with the type of food waste feedstock. As the HTC temperature varied from 180 to 220 °C, the hydrochars produced from taro, lettuce, and watermelon peel varied in the range 2.8 and 12.7%, 0.7 and 1.0%, and 1.7 and 2.2%, respectively.

The effect of temperature on hydrochar’s surface area has also been reported in some studies. As the temperature varied from 200–300 °C, the BET surface area of hydrochar produced from food waste (comprising a mixture of vegetables, fruits, staple foods, and meat waste) varied between 5.23 and 7.14 m^2^/g [63]. Generally, hydrochars produced from food waste were found to have a low surface area (1.76 to 30.59 m^2^/g) and were not suitable for use as supercapacitors, electrocatalysts, and adsorbents without chemical modification of their structure. However, the surface area of food waste-derived hydrochars may be too low to be considered for adsorptive application; this may be compensated by the presence of functional groups. Just like the mass yield, the effect of temperature on surface area has been found to depend on the type of feedstock used. Liu et al. [40] found that as the HTC temperature varied from 180 to 240 °C, the specific surface area of hydrochar produced from lettuce, taro, and watermelon peel varied in the range between 3.67 and 6.90, 0.68 and 9.23, and 3.29 and 8.45 m^2^/g, respectively. Furthermore, an increase from 180 to 220 °C for lettuce waste hydrochars did not significantly affect their specific surface up to 220 °C, beyond which a rapid decrease was observed at HTC temperature of 240 °C. At 220 °C, watermelon peel-derived hydrochar had the highest specific surface area at 8.45 m^2^/g, with stable mesoporous carbon among the three hydrochars produced.

The effect of temperature in most studies on HTC of food waste is also seen on the amount of volatile matter. During the HTC of kitchen waste, it was observed that a rise in temperature led to a decrease in volatile matter accompanied by a corresponding increase in fixed carbon and ash [57]. It is hypothesized that the extent of changes in volatile content of hydochar with increasing temperature is also a function of the type of feedstock [60].

In most studies, increasing temperature generally resulted in a higher carbon content of hydrochar at a given reaction time. A study was carried out for the HTC of food waste at different temperatures (195, 225, and 255 °C) for 12 h. The carbon content of hydrochar increased from 67.72 to 72.99% with increases in the temperature [55]. In another study involving HTC of real cooked FW (without addition of water) at a fixed reaction time (5 h), approximately 10% rise in the carbon content was observed for every 20 °C rise in reaction temperature [10]. In a similar study, as the temperature increased from 200 to 250 °C, the fixed carbon content of food waste increased from 30.34 to 45.41% and subsequently to 47.43% as the temperature was further raised to 300 °C [63]. A similar trend was also observed elsewhere [60].

The effect of temperature on both carbon and oxygen contents was also found to be dependent on the type of food waste. This is because different food wastes possess dissimilar chemical compositions, rates of dissolution, and thermal stability. According to Zhang et al. [65], HTC of lignin-rich food waste results in hydrochar with high carbon content as most of the carbon is retained in it. Thus, appropriate temperatures for particular food waste feedstock should be selected based on the specific application.

The effect of temperature on hydrochar’s oxygen content was also investigated. A study varied the HTC temperature of a complex food waste mixture between 200 and 260 °C at a fixed time of 1 h and observed that the oxygen content of the produced hydrochars reduced from 12.76 to 9.87% with increasing temperature [41]. Liu et al. [40] investigated the impact of temperature parameters on both carbon and oxygen contents of hydrochar produced from lettuce, watermelon peel, and taro waste. It was found that the carbon content of the food waste hydrochar increased while the oxygen content declined with the rising HTC temperature from 180 to 240 °C. An obvious change of the carbon and oxygen contents in the lettuce and watermelon peel hydrochars was observed when the HTC temperature was raised to 240 °C, while this change appeared at a lower HTC temperature (200 °C) for taro waste hydrochar. 

Some studies on HTC of food waste have revealed that an increase in temperature has affected the nitrogen and hydrogen contents to a certain extent, depending on the feed. A study on HTC of food waste with the temperature range between 180 and 220 °C led to a significant reduction in percentage nitrogen content (from 52.76 to 40.97%) [48]. As regards hydrogen, some of the studies revealed that its content in the hydrochar is not affect by temperature [60]. In studies where a noticeable change was observed, no specific trend was noted. However in all the studies, the hydrogen content of the hydrochar improved relatively to the raw food waste feedstock [58].

A positive effect of temperature on heating value of the hydrochar has been observed in many studies. At a fixed reaction time of 5 h, the hydrochars produced at 160 °C had a HHV value of 23 MJ/kg, which increased to 29.5 MJ/kg with a rise in temperature to 180 °C. A further increase in temperature to 200 °C did not show an appreciable increase in HHV [10]. Another study on food waste, carried out at a reaction temperature between 200 and 300 °C, revealed an increase in HHV from 20.81 to 28.98 MJ/kg when the temperature varied from 200 to 250 °C, and a further rise to 31 MJ/kg at 300 °C [63]. A similar finding was observed in another study where there was a consistent increase in HHV with increasing temperature [41].

Research has also shown that temperature also affects the energy density of the food waste-derived hydrochar. The energy density of hydrochar is usually a function of the relative amount of lignin in the corresponding food waste. The HTC of food waste were studied between 200 and 300 °C. It was revealed that the energy density ranged from 20.81 to 31 MJ/kg [63]. Similar findings were observed in other studies [2,10]. It was generally observed that higher HTC temperature caused no significant improvement on the energy densification though an improvement in the carbon densification. Moreover, the effect of varying the HTC temperature for coffee silver skins between 180 and 260 °C significantly led to an increase in the energy density from 1.3 to 1.62 MJ/cm^3^ [66].

Some studies have investigated the effect of temperature on energy yield of the hydrochar. The energy yield of a hydrochar is the product of the mass yield and the energy densification ratio (ratio of the lower heating values (LHV) of the char and feedstock) [67]. A study observed that varying the temperature in a range between 175 and 250 °C gave a maximum energy yield of 63.5% on dry basis at 250 °C [59], consistent with another study where an increase in temperature from 200 to 300 °C led to a corresponding drop in the energy yield from 12.81 to 10.92% on dry basis [63]. However in another study, varying HTC temperature of food waste comprising cellulose (3.12%), hemicelluloses (22.76%), carbohydrates (34.21%), lignin (2.68%), protein (23.28%), and lipids (25.79%) in a range between 210 and 270 °C was found to result in a decrease in energy yield from 56.31 to 20.48% [31]. This was due to higher hydrochar yield and lower energy densification at 210 °C.

Some works reported the effect of temperature on the ease of combustibility of hydrochars. Although increasing the temperature of food waste HTC in some studies did not affect the number of degradation stages, the activation energies were affected. The activation energies for the first and second degradation stages were 56.78 and 47.38 kJ/mol, respectively, for hydrochar produced at 200 °C, while the activation energies of 33.60 and 29.80 kJ/mol were respectively observed at 300 °C [63]. From these results it could be concluded that the activation energy during both stages (first and second) decreased with an increase in temperature from 200 to 300 °C. However, during the HTC of sweet potato waste, an increase in reaction temperature from 180 to 300 °C at 1 h resulted in an increase in activation energy from 197.69 to 264.28 kJ/mol during the first stage and an increase in activation energy from 150.57 to 190.52 kJ/mol during the second stage [68]. The decrease in activation energy during the first stage (which commenced within the temperature range: 315–390 °C) could be explained by the fact that unstable volatile components were released, thereby reducing the specific surface area and porosity of the hydrochar. On the other hand, the decrease in activation energy during the second stage (which commenced within the temperature range: 400–540 °C) could be explained by the occurrence of an aromatization reaction which leads to an increase in aromatic compounds generation.

### 5.3. Contact Time

Contact time also affects the physicochemical properties such as carbon and hydrogen contents, energy density, and mass yield of a hydrochar. Generally, increased contact time of HTC usually led to an enhanced carbon content and reduced mass yield of the produced hydrochars. Several experiments have studied the effect of contact time on HTC of food waste using a time that ranged between 0.5 and 12 h. At very high temperatures, contact time may not have a significant impact on the mass yield except that higher contact time could lead to a complete decomposition of the food waste. Moreover, increasing the contact time may affect the chemical composition of the resulting hydrochar. A study was carried out which revealed that during longer contact time durations, the carbonization of household waste increased the intensity of polymerization and formation of phenolic compounds while decreasing carbonyl compounds formation [56]. The effect of contact time on mass yield was studied for HTC of food waste by changing contact time from 20 to 120 min, where a gradual decrease in hydrochar mass yield from 68.5 to 65.3% was observed [62]. Another effect of contact time on mass yield was found in the HTC of food waste under varying contact time between 5 and 120 min, where a 13% increase in the mass yield was recorded after 60 min [59].

The effect of contact time (1.5–12 h) on the carbon content was studied during HTC of kitchen waste at 225 °C. As the contact time varied from 1.5 to 9.0 h, the carbon content of hydrochar slightly increased from 69.31 to 70.91% [55]. Atallah et al. [4] studied the effect of contact time on HTC of spent mushroom compost waste at 250 °C. A 23.45% increase in carbon content was recorded within 2–4 h time range. In another study, contact time of less than 1 h did not show a significant increase in carbon content of the hydrochar at a fixed temperature of 200 °C. However, after 30 min of contact time, a decreasing trend was first observed. Thereafter, from 1 to 5 h, the carbon content of hydrochar increased in range: 16–20% [10]. Some studies have found that the effect of content on hydrogen and fuel ratio of the hydrochar were not very significant, possibly due to reduced mass yield [10,57], even though Gupta et al. [10] found that increasing contact time from 0.5 to 1 h led to a significant increase in H/C ratio.

### 5.4. Water–Biomass Ratio

Water–biomass ratio is an important process parameter for HTC of food wastes because it determines the extent of hydrolysis reaction or the extent of solubilization of the food waste. Thus, appropriate water–biomass ratio is necessary so as not to cause excessive hydrolysis of the food waste which would result in low mass yield. In addition, the study of the effect of the water–biomass ratio is critical in ensuring an appropriate food waste biomass dispersion in the HTC reactor. According to Guo et al. [69], a low water–biomass ratio will reduce the contact time required for the HTC of the food, which will in turn causes an early onset of the polymerization reaction, thereby leading to higher hydrochar yield.

Limited works have reported effect of water–biomass ratio on the mass yield of hydrochar produced from real food waste. The effect of water–biomass ratio on mass yield of HC produced from sewage was investigated by Xie et al. [62]. It was found that an increase in the water–biomass ratio from 2 to 25 wt.% led to an increase in mass yield from 62.9 to 66.1%. In the HTC of starch and glucose, the water–biomass ratio of the reaction was studied between the ratios of 5 and 10. An increase in the water–biomass ratio from 5 up to 8 led to an increase in the mass yield of hydrochar, whereas decreased mass yield was found at a water–biomass ratio below 5 [70]. In stirred HTC reactors, it is pertinent to regulate the water–biomass ratio in such a way that the efficiency of the stirrer is not retarded. However, some operational challenges were encountered when a water–biomass ratio of 5 and 8 was used during the HTC of acacia wood in a stirred HTC reactor at 200 °C [71]. It was concluded in the study that a water–biomass ratio of 10 led to stirring the solution with ease. Optimization studies need to focus optimization methods in order to ascertain the effect of water–biomass ratio for real and multicomponent food waste as only few studies are available. For example, in the HTC optimization of oat husk using a response surface methodology, the optimal water–biomass ratio of 12.5 was necessary if a hydrochar was to be produced with mass yield of 53.8% with a heating value of 21.5 MJ/kg [72].

### 5.5. Pressure

Few works have analyzed the effect of HTC pressure on the properties of food waste-derived hydrochar. This may be due to the fact that the pressure of an HTC reactor in most studies [73,74] were measured or monitored after few minutes into the duration of dwelling time. According to Das et al. [75], HTC process normally records a final pressure of about 2 MPa at 200 °C within 1 h of dwelling time. Lachos-perez et al. [76] opined that pressure may not be a key parameter in HTC of food waste, especially in batch reactors. In continuous reactors, a pressure of 300 bar was reported to show significant intense graphitization of hydrochar leading to improved heating values, however, very high pressure can lead to high operating costs. However, increased pressure is important because it increases the hydrolysis and decomposition rate of the food waste. In addition, pressure is required to keep the aqueous reaction media in a subcritical state and expedite the exchange of hydrogen ions.

### 5.6. Heating Rate

The effect of heating rate on the properties of hydrochar is sparse in the literature. Most studies were carried out at heating rates ranging between 2 and 15 °C/min. The heating rate is critical when controlling the distribution of the solid, liquid, and gaseous products of the HTC of food waste. Higher heating rate is known to result in very low hydrochar yield. However, higher heating rate may reduce heat transfer restrictions and prevent a secondary reaction that may lead to stable products [68]. Lower heating rate increases the carbon content of the char with reasonably low calorific value.

### 5.7. Properties of the Produced Hydrochars

Hydrochars have specific properties which qualify them for different applications. Many studies have shown that the chemical and energy properties of a hydrochar are largely dependent on both the process parameters and the type of food waste used. The use of hydrochar for different applications is dependent upon its elemental composition, ultimate composition, and energy properties. Table 3 [2,28,40,51,55,56,59,77,78,79,80,81] summarized elemental composition, ultimate composition, and energy properties of hydrochars produced from different types of food waste materials.

The volatile matter of food waste ranged between 50 and 65.4%, except for chicken and household wastes where it is greater than 90%. The fixed carbon content of food waste that hydrochar produced was in the range of 31.1 to 49.3%, except for corn fiber, household waste, and chicken, which were less than 10%. Cooked food and protein-rich food waste was found to contain less ash than fruit peels, which in turn was also less that cereal waste. The ash content was also less than the value of 20 to 40% that is normally contained in lignite coal. The ash content present in the hydrochar comes from the alkali metals (Na and K) in corresponding food waste, and the amount present in the hydrochar determines its application as a biofuel. The alkali metals (Na and K) resulted in fouling of the heating surfaces of the boilers, thereby reducing their thermal efficiency.

Elemental composition of hydrochars produced from different food wastes shows that the carbon contents ranged between 45 and 71%, while the hydrogen contents are less than 10%. The relatively higher elemental carbon content in food waste compared to a carbon content range of 45–51% in coal may be due to the higher carbohydrate content of food waste. Except for simulated waste, chicken, and brewers spent grain hydrochar, which had nitrogen content above 5.0%, most of the food wastes ranged between 1.5 and 3.74%. It is worth noting that hydrochar developed from pomegranate residue grape marc had a nitrogen content less than the value of 1.82% contained in lignite [82]. This signifies that these fruit waste-derived hydrochars may be a more appealing source of clean energy. The sulphur contents of most of the food wastes were less than 0.6% (found in Canadian coal). Except for grape marc hydrochar, which had an oxygen content of about 40%, most food waste had an oxygen content of less than 37%, which is found in Thailand coal. These results show that food waste hydrochars could be a potential co-feed with coal in combustion plants for energy generation.

In terms of energy properties, most studies showed that the heating values of food waste hydrochars are greater than 20 MJ/kg. This suggests that food waste hydrochars are commercially feasible for use as biofuels [2]. The high heating content of the hydrochar developed from food waste is comparable with coal (15 to 35 MJ/kg) [83]. This might be due to their high lignocellulosic content. Usually, according to ISO/TS 17225–8, a hydrochar with heating value between 18.6 and 26.2 MJ/kg would be allowed for its application as a biofuel in industry [60]. The fuel ratio of most food wastes is ranged between 0.62 and 0.98 except for chicken, household waste, and corn fiber, which have a fuel ratio less than 0.2. A hydrochar with a fuel ratio of less than 2.5 meets the standard for use as solid fuel in pulverized combustion systems [2,59]. Some studies have investigated the energy yield of food waste hydrochars from fruit peels, and the values ranged between 60 and 70%. The energy yield for corn fiber is found to be much lower (27.26%) (Table 3).

### 5.8. Combustion Kinetics of the Produced Char from HTC

Combustion of hydrochars is carried out by heating the samples from room temperature to a high temperature (>1000 °C) under air flow at a particular heating rate. Using a DTG curve, the combustion process may be divided into three stages depicting the devolatilization, combustion phase, and the burnout phase. A thermogravimetric analyzer is usually used to study the combustion kinetics of hydrochar, after which activation energy is determined to evaluate the process since it essentially affects the temperature sensitivity of the combustion rate. The thermogravimetric analysis (TGA) parameters reported for combustion kinetics studies of hydrochar produced from different food wastes are presented in Table 4 [31,56,68,84,85,86,87,88]. The combustion kinetics experiments were conducted using a thermogravimetric analyzer in order to determine the characteristic of the hydrochars. Combustion kinetics analysis of the hydrochar mass loss at different time and temperature intervals usually provides insight into its combustion behavior as well as the necessary data required to design large-scale combustion plants. The combustion kinetics analysis of the food wastes (Table 4) is carried out in the temperature range between 25 and 1000 °C under air flow at the volumetric flow rate ranging between 16 and 100 mL/min. The heating rate in most of the studies is maintained between 5 and 40 °C/min. Generally, most of the studies used a sample weight in the range between 5 and 15 mg to improve the flow of heat between the wall of the crucible and sample.

Most of the combustion kinetics studies presented in Table 4 limited their studies to only the combustion behavior of food waste [31,86] (Sharma et al., 2022; Su et al., 2021) and did not study the kinetics. For example, Su et al. [31] carried out a TGA of hydrochar. Results showed that compared to oil extracted food waste, the hydrochar had a more stable and longer combustion process with the higher ignition temperature and burnout temperature. Likewise, Sharma et al. [86], found that hydrochar produced from a mixture of food and yard wastes exhibited a smoother combustion profile with a single combustion zone which depicts its improved combustion stability. A further kinetics analysis of the combustion of the hydrochar produced in these studies under reference would deepen an understanding of complex reaction in addition to providing a basis for cost-effective and eco-friendly conversions of these hydrochars in the future. Table 5 displayed the combustion kinetics parameters (activation energy, pre-exponential factor, and reaction order) obtained for some hydrochars using two common model-free kinetic approaches of Kissenger–Akahira–Sunose (KAS) and Flynn–Wall–Ozawa (FWO). These approaches requires the determination of kinetics parameters from the TGA data obtained at different heating rates for a given sample [84,85,89]. These methods were usually reported in most of the studies probably because of their simplicity or for the need to avoid mistakes that are commonly associated with choosing a particular reaction model. Noteworthy in Table 5 [68,85,87,88] is the decreasing trend of the activation energy kinetics parameter of the hydrochars across the different stages (2 to 4) in the combustion. Islam et al. [89] investigated the combustion kinetics of hydrochar produced from Karanj fruit hulls using the TGA data obtained at three different heating rates between 5 and 20 °C/min using the KAS. During the study, the activation energy varied from 114 to 67 kJ/mol as the conversion varied from 0.1 to 0.8. Kojić et al. [85] investigated the combustion kinetics analysis of hydrochar produced from spent mushroom substrate at 180 °C and found that E_a_ decreased from 158.83 to 38.25 kJ/mol (FWO) and from 157.64 to 27.89 kJ/mol (KAS). Combustion kinetics analysis of the hydrochar produced from the same substrate at 260 °C also showed a decrease in E_a_ from 133.06–64.25 kJ/mol (FWO) and from 130.18–55.35 kJ/mol (KAS). A similar decreasing trend was also observed during the combustion kinetics analysis of mixed food waste [88] using the classical Arrhenius equation where respective activation energies calculated for the first and second peaks were 25.47 and 16.52 kJ/mol. The decrease in activation energy as the combustion reaction progressed was expected given that the cellulose and hemicellulose structure of the hydrochar would be destroyed at higher temperatures. Apart from this factor, the observed decrease in activation energy may be attributed to the improved surface area and porosity of the hydrochar. As presented in Table 5, the values of the activation energy for different hydrochars were different depending on the type, complexity, and combustion characteristics of the food waste. Also, it would be noted that the values of activation energies obtained in most of these studies were <179 kJ/mol, reported for low-rank coal [63]. The results suggest that the hydrochar produced from food waste could be a potential substitute for conventional fuel.

Combustion characteristics of raw lignite and the corresponding hydrochar were studied by thermogravimetry coupled with differential scanning calorimetry (TG-DSC) by heating the sample from room temperature to 700 °C under atmospheric air [90]. The activation energy of the hydrochar samples increased with temperature increasing at char combustion stage. It was found out that the hydrochar produced at the temperature of 230 °C required a lower activation energy for combustion (55.36 kJ/mol) compared to the higher temperature used. In addition, the hydrochar produced at 230 °C had activation energy lower than raw lignite (57.38 kJ/mol). High activation energy indicated that the hydrochar samples upgraded at high temperature were difficult to combust.

## 6. Future Research Directions

This review has shown some gaps in research on HTC of food that need to be filled. Although HTC parameters such as temperature, contact time, water–biomass ratio, and pressure affect the physicochemical properties of hydrochars, future research should focus on investigating the effect of using different acid catalysts and concentrations (for catalytic HTC of food waste). A suitable catalyst for HTC of food waste should significantly improve the hydrolysis level of the food waste. Moreover, a good criterion for choosing a potential catalyst should encompass effectiveness, cost-effectiveness, and high selectivity towards required yield. In this way, the yield and composition of the char could be enhanced. Furthermore, the effect of heating rate and water–biomass ratio should also be investigated using Design of Experiments (DOE) approach. In addition, the effect of particle size on the physicochemical properties of hydrochar should also be examined. Finally, the future investigations should also focus on identification of a suitable reactor for the hydrothermal treatment process of food waste.

## 7. Conclusions

Recent developments in HTC of food waste to carbonaceous solid fuel were reviewed. The effect of temperature on both carbon and oxygen contents of hydrochar was found to be dependent on the type of food waste. The fixed carbon contents of food waste hydrochar produced were in the range of 31.1 to 49.3%, except for corn fiber, household waste, and chicken meat, which were less than 10%. Cooked food and protein-rich food wastes were found to contain less ash content than fruit peels, which in turn was also less that cereal waste. The ash content was also less than the value of 20 to 40% that is normally contained in lignite coal. These results show that hydrochars produced from food waste could be a suitable substitute for conventional fossil fuels.

## Figures and Tables

**Figure 1 foods-11-04036-f001:**
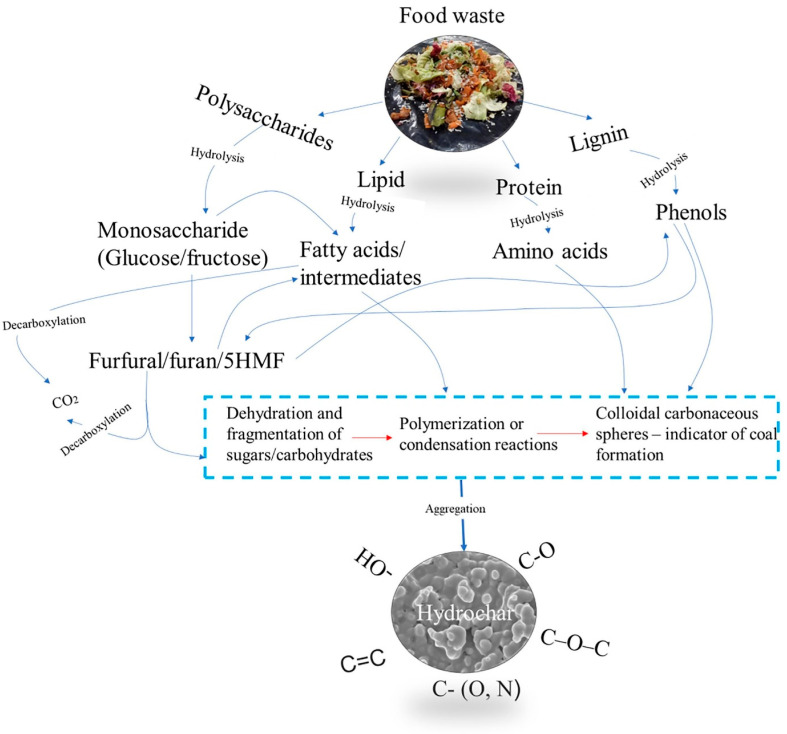
Reaction pathway for the HTC of food waste into hydrochar [26].

**Figure 2 foods-11-04036-f002:**
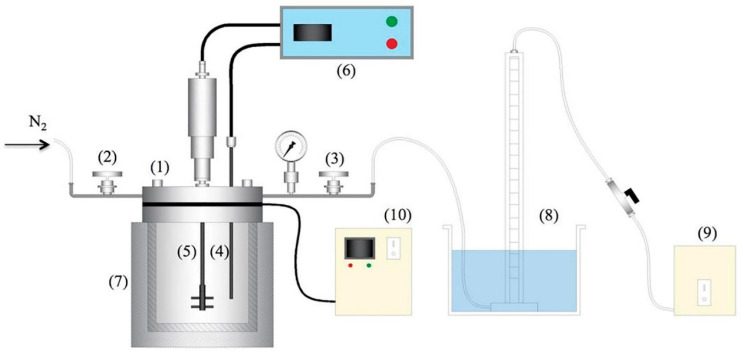
Reactor set-up of a typical HTC system (**1**) 500 mL reactor; (**2**) inlet valve; (**3**) outlet valve; (**4**) thermocouple; (**5**) stirrer; (**6**) controller; (**7**) tubular furnace; (**8**) gas measure system; (**9**) vacuum pump; (**10**) chiller [51].

**Table 1 foods-11-04036-t001:** Characteristic of previously reported food wastes (dry basis).

Category	Food Waste	C (%)	HC (%)	L (%)	Moisture Content (%. wet)	Ash (%)	HHV (MJ/kg)	O/C	Fuel Ratio (FC/VM)	Reference
**-**	Mixed	-	-	-	62.2	5.41	19.76	1.02	1.12	[24]
**-**	Mixed	2.0	1.2	0.1	3.30	10.3	-	0.65	0.21	[25]
**-**	Mixed	-	-	-	-	5.68	10.54	1.06	0.21	[26]
**Plant**	Apple	-	-	-	-	2.30	-	0.97	0.23	[27]
**Animal-based**	Chicken	-	-	-	-	2.37	25.32	0.41	0.27	[28]
**Vegetal**	Cabbage	-	-	-	-	2.10	17.77	0.89	0.12	[28]
**Carbohydrate-rich**	Rice	-	-	-	-	0.29	18.33	1.05	0.16	[28]
**Vegetal**	Mixed	-	-	-	-	11.4	16.7	1.00	0.014	[20]
**-**	Mixed	-	-	-	74.0	1.15	22.74	0.71	-	[29]
**-**	Mixed	36.63	1.12	15.61	9.60	3.62	16.07	0.98	0.17	[30]
**-**	De-Oiled food	3.12	22.76	2.68	1.59	13.01	19.16	0.548	0.064	[31]

- = Not reported, C = cellulose, HC = Hemicellulose, L = Lignin; HHV = High heating value; FC = Fixed carbon; VM = Volatile matter.

**Table 2 foods-11-04036-t002:** Properties of hydrochar developed from HTC of food waste under optimum conditions.

**Food Substrates Components**	**HTC Optimum Conditions**	**Key Property of Hydro Char Targeted**	**Value**	**Reference**
Lettuce/taro/watermelon peel	Lettuce (180–240 °C) 180 rpm for 3 h	2,4-D adsorption	88.4. mg/g	[40]
Discarded vegetables and meats, potatoes and less fruit peels and eggshells.	225 °C, 4.5 h	Methane yield	19%	[55]
Cooked meat, vegetables, rice, noodles, fruit peels, vegetable parts, and condiments, paper cups, and woody chopsticks	230–260 °C, 8 h	Compressive strength Impact resistance index	2.37 MPa 10	[41]
Mainly of fruit and vegetables	200 °C, 1 h	H/C O/C	1.41 0.52	[2]
Cooked FW (as received) without addition of water	200 °C, 1 h, 2 L	Heating value	~30 MJ/kg	[10]
Cooked rice, chicken, fruit and vegetable peels, lentils, and bread	200 °C, 1–8 h, 2 L	Heating value	~27 MJ/kg	[56]
Household kitchen waste	300 °C. 1.25 h, 0.5 L 35 MPa	Heating value	20.63 MJ/kg	[57]
Food waste (51.4 wt.% carbohydrates, 15.7 wt.% lipids, and 27.5 wt.% proteins)	180–220 °C, 0.25–0.5 h	Fatty acid retention Net fat recovery	78% ~50%	[58]
Real kitchen waste	260 °C, 1 h, 0.5 L, 4 °C/min, 100 rpm	Ammonium concentration	929.75 mg/L	[48]
Municipality food waste	200–300 °C, 1 h, 1 L 30 bar, N2 gas, 600 rpm.	Carbon content Heating value	39–73% 15–31 MJ/kg	[59]
Defrosted feedstock	170–230 °C,1 h, 4 L 1.5 kg, 3 °C/min	Heating value Ash Fixed Carbon	18.6–26.2 MJ/kg <7.0% <45%	[60]
Retail-level food waste	250 °C, 1 h, 1 L 0.55–0.58 MPa, 400 rpm.	Hydrochar partitionability Best solvent	50% Ethanol	[61]

**Table 3 foods-11-04036-t003:** Chemical compositions of hydrochars produced from different types of food waste materials (dry and ash-free basis).

Waste	Sample	Proximate Composition			Ultimate Composition (wt.%)					Energy Properties			Reference
		VM	FC	Ash	C	H	N	S	O	**HHV** (MJ/kg)	**Fuel Ratio**	**EY** **(%)**	
Raw food waste	HC-230	56.2	29.5	14.3	54.8	6.1	2.3	0.2	23.7	23.7	-	-	[2]
Municipal waste	HC-200	51.0	43.6	5.4	58.21	5.15	3.01	0.14	28.08	23.3	-	57.1	[59]
Lettuce waste	HC-220	-	-	8.7	63.3	7.21	3.41	-	26.1	-	-	-	[40]
Watermelon peel	HC-220	-	-	2.5	62.3	6.26	3.09	-	28.4	-	-	-	
Taro	HC-220	-	-	0.5	68.6	5.30	2.09	-	24.0	-	-	-	
Pineapple peel waste	HC-200	59.4	38.9	1.7	61.1	5.3	-	-	30.9	25.1	0.65	62.8	[51]
Orange peel waste	HC-200	60.8	37.5	1.7	60.7	5.2	-	-	31.3	24.8	0.62	63.3	
Tangerine peel waste	HC-220	59.5	38.4	2.1	61.6	5.3	-	-	29.3	25.5	0.65	60.7	
Corn Fibre	HC-220	65.40	7.84	2.53	65.40	7.84	2.53	0.21	23.78	-	0.12	27.26	[77]
Pomegranate residue	HC-220	-	-	-	56.14	6.06	1.54	0.45	35.85	21.27	-	68.79	[78]
Brewer’s spent grain	HC-220	64.04	31.10	4.86	65.90	6.48	5.13	0.09	17.54	27.04	0.49		[79]
Grape marc	HC-220	58.3	39.0	2.7	51.7	6.5	1.5	-	40.3	21.3	0.69	-	[80]
Kitchen waste	HC-225	-	-	1.22	70.98	7.05	3.74	-	16.53	32.19	-	-	[55]
Household wet waste	HC-200	91.4	5.9	2.7	58.4	6.4	2.8	-	29.7	22.7	0.06		[56]
Simulated food waste	HC-220	56.4	38.4	5.2	60.9	5.2	6.0	-	22.7	-	0.68	-	[81]
Cabbage (raw)	HC-220	61.2	38.5	0.32	62.8	5.33	2.95	0.42	28.18	25.28	0.62	-	[28]
Rice (cooked/dried)	HC-220	50.2	49.3	0.52	65.6	4.9	2.03	0.55	26.4	25.96	0.98	-	
Chicken	HC-220	94.8	4.2	1.02	66.1	9.9	6.46	0.3	16.22	32.97	0.04	-	

C—Carbon; H—Hydrogen; N—Nitrogen; S—Sulphur; O—Oxygen; HHV—High heating value.

**Table 4 foods-11-04036-t004:** TGA parameters reported for kinetic studies of combustion of hydrochar for different food waste.

Food Waste Substrate	TGA Reactor Model	Sample Weight (mg)	Temp. Range (°C)	β (°C/min)	Φ (mL/min)	Reference
Beet pulp	Netzsch STA 449 F3 Jupiter	10	≤700	10, 20, 30	40	[84]
Spent mushroom	Setaram Setsys Evolution 1750	7	25–1000	5, 10, 20	16	[85]
Sweet potato	TGA, METTLER TOLEDO	8 ± 0.5	100–800	20	100	[68]
Oil extracted food waste	Discovery SDT 650	-	25–950	10	100	[31]
Food/yard waste	Perkin Elmer Pyris Diamond	-	≤900	12	100	[86]
PVC and bagasse		-	30–900	20	100	[87]
Mixed food waste	STA 449 F5 Jupiter	10 ± 0.5	50–900	10	-	[88]
Household wet waste	Shimadzu DTG-60 TGA	10–15	≤950	10	100	[56]

Φ = air flow rate, β = heating rate.

**Table 5 foods-11-04036-t005:** Kinetic parameters of combustion of hydrochar from food waste.

Feedstock	Hydrochar Code	Kinetic Modelling Approach	Stages	Activation Energy, E (kJ/mol)	Reaction Order, n	Frequency Factor, A (1/s)	Reference
Mixed food waste	HTC220	Arrhenius	2	Stage 1: 25.47 Stage 2: 16.52	3.0 0.9	11.36 × 10^−2^ 7.37 × 10^−2^	[88]
PVC_R_ was co-treated with bagasse	HC-P-S	-	2	Stage 1: 86.07 Stage 2: 47.62	-	-	[87]
Spent mushroom substrate	SMS-180/260	Flynn-Wall-Ozawa (FWO) and Kissinger-Akahira-Sunose (KAS)	3	81.76 (FWO) and 75.22 (KAS) for SMS180 and 91.99 (FWO) and 85.71 (KAS) for SMS260	-	-	[85]
Sweet potato	220–60	Coats–Redfern integral	2	Stage 1: 211.94 Stage 2: 181.65		7.58 × 10^17^ 2.40 × 10^9^	[68]

## Data Availability

Data is contained with this article.

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
