# Peer review of "Hydrothermal Conversion of Food Waste to Carbonaceous Solid Fuel—A Review of Recent Developments"

_foods, 2022, doi:10.3390/foods11244036_

Round 1

Reviewer 1 Report

This was an interesting review on an interesting issue.

Here are my few comments:

There is no need for list of contents.

Line 63: One of the SDGs goal is to cut food loss and waste to half by 2030, while you just mention to food loss. The reference should be changed, accordingly.

Please describe the methodology of this review.

Author Response

File attached

Reviewer 2 Report

It looks mere compilation of the available information with the known facts. Author contribution needs to be highlighted. It lacks the motivational/ innovative approach for the readers. 

Author Response

File attached

Reviewer 3 Report

The review discusses recent development in HTC of food waste and its conversion into solid fuel. The discussion cover food waste properties, the fundaments of the HTC reactor, influence of the HTC process parameters (temperature, process time, pressure water to biomass ratio, and heating ratio) on the properties of hydrochar made from food waste. The title is informative and relevant to the content of the work. The study aim is clearly presented and used references are recent and relevant to the topic.

             The introduction briefly summarizes what is already known about HTC of food waste, what can we expect from the process, and what other researchers did in the field of HTC. The research question is clearly outlined and information presented in the Introduction justifies the research need for a review. Nevertheless, there is still place for improvement:

·         Since it is a review paper it would be good to add information that shows the wide perspective to the reader: how much food waste is produced worldwide/ per person etc., what are the types of food waste edible/inedible, avoidable/unavoidable, and what are current methods of dealing with them what are the pros and cons of these methods. It needs to be shown to which kind of waste is reasonable to use HTC for solid carbonaceous fuel production and which is better to use other methods

·         Line 63-64 – Is this plan of the UN for some specific region or the whole world?

·         Line 76-77 – how does dielectric constant affect the process and why is important to mention it here? I didn’t see an explanation for this constant in other parts of the manuscript.

·         Line 91-93 – what techniques do you mean? please describe them briefly.

There is a lack of a methods section. Though this is a review paper, the methodology should be given about how work was done, what kind of articles were used, how they were found, etc., There also can be presented the structure of the article, that readers know what to expect from the specific part of work.

In the manuscript, characteristics of food waste and hydrochars are compared (Chapter 2.2) but no section/chapter explains what these parameters meant and why they are important when solid fuels are analyzed/compared. The same situation is when the HTC process is described. I suggest adding such kind of information as a separate chapter so readers not familiar with solid fuels and thermal conversion were able to understand the article fully, especially since the journal is about food science, not energy.

            The data presented in Table 1 is unclear.  The authors show the results from different analyses without explaining what is in the Table and why it was placed there together. There are results of (i) cellulose, hemicellulose, and lignin determination, (ii) (part of proximate analyses) (iii)part of elemental analyses, and ()IV analyses of calorific value. These results are not agglomerated in a typical way. Usually, proximate and ultimate analyses are given together in the full spectrum while here we see only chosen parameters and their derivatives like O/C or fuel ratio. What is the reason for that? Why does this table differ so much from Table 3 which construction is correct?  Please consider the improvement of Table 1.

            Table 1 and Table 3 should state what base results are presented (wet, air-dry, dry, dry ash-free). Currently is hard to guess to compare results.

·         Line 176-177 – Please check and provide references for the statement that high moisture content in food waste causes the release of dioxins when burnt. According to my knowledge dioxins are produced always when solid fuel is burnt in improper conditions as products of not fully combustion when chlorine compounds are available. Therefore, I’m not sure if the water content is directly responsible for dioxin release. 

·         Line 184 – many types of coals exist and differ in calorific value, ash content, and moisture content. Please check more research about coal and improve comparison.

·         Lines 191–192 Please clarify what is the reason of what. The moisture affects O/C, or O/C affects moisture?

·         Line 198 – What are the combustibility index and volatile ignitability? Why do they have the same unit as HHV?

·         Line 218 – What kind of reactivity do you mean? In the next line (219), you state that “High volatile matter (VM) indicates that the fuel can be easily ignited and subsequently oxidized” As far as I understand this means that the more volatile matter the more material is reactive because it is easier to ignite. So this is kind of opposite to the statement from line 218.

·         Section 3.1.2. It would be good to provide information about the range for sub- and super-critical conditions, differences in process performance, and product quality. Also, it would be good to explain why landfill leachate was used as a reactant medium. Maybe here you can discuss also the effect and role of dielectric constant?

·         Section 3.1.3. There is a lack of references. Only Figures have references. Does it mean that the whole section and HTC process reaction description were done using only those two references?  A review article should have more relevant references.

·         Line 305-308 Please provide references. Are you sure that glass reactors are used for the HTC process? During HTC pressure can grow high, I am not certain that such a reactor can hold it.

·         Lines 316-317 Please provide references to those studies

·         Lines 319-322 need clarification. The term “Hydrolysis temperature” is here used for the first time. Please explain this term. Maybe it would be good to add this information to section 3.1.1.” HTC reactor pre-heating and reaction time”

·         Figure 3. It would be good to present also two-stage reactor and the basic principles of how it works

·         Lines 346-347 need clarification. Why does the large volume required make HTC reactor attractive?

·         Line 361 needs clarification/additional explanation. The review is about carbonaceous solid fuel while in Table 2, other properties are also described e.g. Adsorption, Methane yield, etc.

·         Lines 379-385, please provide references

·         Lines 386-387, please provide references

·         Lines 438-439 Please check numbers 2.76 and 9.87. The text says that oxygen content was reduced while numbers show an increase.

·         Lines 469-470 Are you sure that energy density is given in %?

·         Lines 476-485 It would be good to add information about what is energy yield, how is calculated, and what it says about the process. More importantly, are these values you compare given on the same bases? From my experience, some authors provide EY results on a wet basis while others do on a dry base. Here we can see quite a big difference between 63.5 and 12.841-10.92 for similar temperatures. Please check it and if needed provide results in the same base

·         Lines 476-485. Apart from energy yield, there are also other properties used to describe the carbonaceous fuel production process. Mass yield, energy densification ratio/enhancement factor, and energy gain. Why you didn’t discuss them as well?

·         Lines 486-503 – It would be good to provide more info about combustion, what stages there are, why activation energy is important and what tell us about the process. Also, it would be good to provide examples of typical solid fuel combustion e.g. coal or wood for comparison.

·         Section 4.1.4. Please provide references

·         Lines 600-609 Please provide references

·         Lines 612-614. Are those compared values given on the same basis? Please provide references to coal.

·         Line 626 Are you sure that coal has only 15-16 kJ/g? Maybe you mean low heating value here? Usually, most of the coals used for electricity generation have an HHV of at least 20 MJ/kg. Also, Please keep the unit uniform, earlier in the manuscript MJ/kg was used.

·         Lines 658-659 in comparison to what, the hydrochar had a more stable and longer combustion process?

·         Table 5. Please provides information about the model/approaches used for kinetic parameters determination for each reference. It would be good to replace the hydrochar code with process parameters used for its production.

·         In the text (line 669) is information that Table 5 contains kinetics parameters of model-free approaches (KAS and FWO). Nevertheless, the first substrate in Table 5 has a reaction order (n). Please check if kinetics was there determined with the free kinetics model approaches.

·         Table 5. Why for spent mushroom substrate there is generalized information about activation energy instead of precise information for each stage?

· Table 5. The frequency factor has two different ways of recording. 

Author Response

File attached

Reviewer 4 Report

This is a well-prepared review with lots of hard work and efforts. I have minor suggestion for this manuscript.

1. Fig1 may be not necessary. The main information of Fig.1 was about the waste could be changed to gas, liquid and solid carbon after the HTC treatment. However, this is common sense. The authors wanted to explain the proportion of C transformed into these three phases. But the information was in the context not in the figure. And the Fig. 1 was not very precise with the words it used. Either to modify or redraw Fig. 1 or to delete Fig. 1 was my suggestion.

2. There are many abbreviations in the Fig. 2 (e.g.HMF), Table 1 (e.g. FC/VM), Table 3 (e.g.HC-200) without annotation. Annotations should be added to these figures and tables.

3. Fig.3 is confusing. What is the tiny small figure in the up-right corner? If that reactor means the same thing with the main reactor, then only one reactor is enough in this figure.

4. As showing in Table 2, most of the HTC reactor was small container with 0.5-1 L volume. Since there were large amount of food waste worldwide, 1 L volume will not fulfill the needs of industrial application. Could authors find same industrial example for the large quantity application? If not, could authors point out what are the shortcomings for the HTC process, especially for the industrial application.  

Author Response

File attached

Round 2

Reviewer 3 Report

The responses were hard to find due to the lack of references to lines in the improved manuscript. For the next time please, provide places where improvements were done.

The manuscript has been revised according to most of the indicated comments.

The authors omitted one comment

Since it is a review paper it would be good to add information that shows the wide perspective to the reader: how much food waste is produced worldwide/ per person etc., what are the types of food waste edible/inedible, avoidable/unavoidable, and what are current methods of dealing with them what are the pros and cons of these methods. It needs to be shown to which kind of waste is reasonable to use HTC for solid carbonaceous fuel production and which is better to use other methods 

Here are my concerns about the author responses to comments.

·         Response to comment 3 – what is the difference between anaerobic digestion and fermentation?

·         Response to comment 5 – About sentence:

A food waste with high lignin content is desired for the production of high yield solid fuel because of its relatively high thermal stability compared to cellulose and hemicellulose”. - What do mean by high-yield solid fuel?

Hitherto, the relatively high moisture content in food waste, especially in fruits and vegetables causes the release of dioxins when burnt with other organic material” - Is there a need for the word Hitherto?

·         Response to comment 9. From my perspective, in the article you produce fuel that is supposed to replace coal, therefore I think that it would be good to show differences for better comparison between produced hydrochars and coals. But I understand your decision.

·         Response to comments 17 – I wasn’t able to find added lines covering the basic principle of the two-stage HTC reaction in Section 3.2.

·         Response to comment 36 – What is the problem to change e.g. 0.113604 to 11.3E-2 or vice versa to keep the same style through the manuscript and Table?

Please reconsider including omitted comment and my concerns to responses.

Best regards  

Author Response

File attached
